# Sodium-Glucose Cotransporter-2 Inhibitors at Discharge from Cardiology Hospitalization Department: Decoding A New Clinical Scenario

**DOI:** 10.3390/jcm9082600

**Published:** 2020-08-11

**Authors:** José Rozado, Daniel García Iglesias, Miguel Soroa, Alejandro Junco-Vicente, Noemí Barja, Antonio Adeba, María Vigil-Escalera, Rut Alvarez, Francisco Torres Saura, Esmeralda Capín, Laura García, María Luisa Rodriguez, David Calvo, Cesar Moris, Elías Delgado, Jesús María de la Hera

**Affiliations:** 1Department of Cardiology, Hospital Universitario Central de Asturias, 33011 Oviedo, Spain; joserozadocast@gmail.com (J.R.); danigarciaiglesias@gmail.com (D.G.I.); miguelsoroa.o@gmail.com (M.S.); ajuncovicente@gmail.com (A.J.-V.); noeminbg@gmail.com (N.B.); antonio.adeba@gmail.com (A.A.); mariavigilescalera@gmail.com (M.V.-E.); rutalvarez3@gmail.com (R.A.); es_capin@hotmail.com (E.C.); laugar1960@hotmail.com (L.G.); mrsjlgc@telefonica.net (M.L.R.); davidcalvo307@gmail.com (D.C.); cesarmoris@gmail.com (C.M.); 2Instituto de Investigación Sanitaria del Principado de Asturias (ISPA), 33011 Oviedo, Spain; eliasdelga@gmail.com; 3Department of Cardiology, Hospital Universitario Vinalopo y Hospital Universitario Torrevieja, 03293 Torrevieja, Spain; ftorressaura@gmail.com; 4Medicine Department, University of Oviedo, 33011 Oviedo, Spain; 5Department of Endocrinology, Hospital Universitario Central de Asturias, 33011 Oviedo, Spain

**Keywords:** cardiology service hospital, glucose-lowering drugs, heart failure hospitalization, hospital discharge, major cardiovascular events, mortality, sodium-glucose cotransporter 2 inhibitors, type 2 diabetes mellitus

## Abstract

Sodium-glucose cotransporter-2 inhibitors (SGLT-2 inhibitors) are new glucose-lowering drugs (GLDs) with demonstrated cardiovascular benefits in patients with heart disease and type-2 diabetes mellitus (T2DM). However, their safety and efficacy when prescribed at hospital discharge are unexplored. This prospective, observational, longitudinal cohort study included 104 consecutive T2DM patients discharged from the cardiology department between April 2018 and February 2019. Patients were classified based on SGLT-2 inhibitor prescription and adjusted by propensity-score matching. The safety outcomes included discontinuation of GLDs; worsening renal function; and renal, hepatic, or metabolic hospitalization. The efficacy outcomes were death from any cause, cardiovascular death, cardiovascular readmission, and combined clinical outcome (cardiovascular death or readmission). The results showed that, the incidence rates of safety outcomes were similar in the SGLT-2 inhibitor or non-SGLT-2 inhibitor groups. Regarding efficacy, the SGLT-2 inhibitors group resulted in a lower rate of combined clinical outcomes (18% vs. 42%; hazard ratio (HR), 0.35; *p* = 0.02), any cause death (0% vs. 24%; HR, 0.79; *p* = 0.001) and cardiovascular death (0% vs. 17%; HR, 0.83; *p* = 0.005). No significant differences were found in cardiovascular readmissions. SGLT-2 inhibitor prescription at hospital discharge in patients with heart disease and T2DM was safe, well tolerated, and associated with a reduction in all-cause and cardiovascular deaths.

## 1. Introduction

Over the last years, a new family of glucose-lowering drugs (GLD), the sodium-glucose cotransporter inhibitors (SGLT-2 inhibitors), with proven cardiovascular benefits [1,2,3,4,5], has shown up. Moreover, their use has been supported by clinical practice guidelines [6,7]. Up till today, the recommendation when prescribing SGLT-2 inhibitors is to mimic the pivotal trial’s clinical scenarios, where the patients were in a stable situation [8,9,10]. Large observational studies consistently showed their efficacy and safety in real-life routine practice [11]. However, there is no such evidence in hospitalized patients, because patients with recent acute heart failure (HF) or acute coronary syndrome were excluded from the clinical trials [8,9,10]. Nevertheless, the early initiation of SGLT-2 inhibitors could become a treatment option due to the size of their effect and the rapid onset of action, especially considering that the greatest risk of cardiovascular events and HF hospitalization is concentrated during the first year after hospital discharge [12]. Although there has been experience with initiation during the hospitalization [13], it seems complicated in terms of hemodynamic stability. Therefore, we suggest it could be tested immediately at clinical discharge, as a new scenario.

Considering the benefit that SGLT-2 inhibitors provide to patients with type 2 diabetes mellitus (T2DM) and known cardiovascular disease, it draws attention to the low prescription rate of these therapies among cardiologists [14] and the lack of scientific evidence for their use at hospital discharge. Hence, it reinforces the urge to study this new clinical scenario. Therefore, the primary aim of the present prospective study is to assess the safety of SGLT-2 inhibitors when prescribed at hospital discharge to T2DM patients after a non-elective admission to the cardiology department. The secondary aim is to assess the cardiovascular efficacy of these drugs in the same scenario.

## 2. Experimental Section

### 2.1. Study Design

This is a single-center, observational, longitudinal cohort study with a prospective follow-up, including consecutive patients with a prior diagnosis of T2DM or new diagnosis of T2DM during hospitalization, who had been discharged from cardiology department at a tertiary care hospital (northwest Spain, southern Europe) between 1 April 2018 and 28 February 2019.

This study (code 36-19) was approved by the Clinical Research Ethics Committee of Asturias on 5 July 2019, met the requirements and standards of the Declaration of Helsinki and its subsequent amendments for research studies in humans, as well as the current data protection regulations in Spain. All participants provided written informed consent.

### 2.2. Patient Population

Eligible patients aged >18 years who were not electively hospitalized in the cardiology department and discharged between 1 April 2018 and 28 February 2019 with diagnosis of T2DM, defined as any one of the following: (i) prior diagnosis of T2DM and at least one or more prescriptions of GLDs at discharge or (ii) new diagnosis of T2DM during hospitalization (according to the American Diabetes Association criteria [15]) and at least one or more prescriptions of GLDs at discharge. The exclusion criteria were as follows: (i) non-DM, (ii) type 1 diabetes mellitus, (iii) T2DM without GLDs at discharge, (iv) T2DM patients treated with SGLT-2 inhibitors during hospitalization, (v) elective hospitalizations, (vi) very brief hospitalization (<3 days), or (vii) inability to comply with follow-up procedures.

### 2.3. Study Procedures

Patients were included in the study before discharge and classified in two groups as follows:SGLT-2 inhibitor group: Patients who had initiated SGLT-2 inhibitors at discharge from the cardiology department (included empagliflozin, dapagliflozin or canagliflozin). These patients could be on other GLDs.Non-SGLT-2 inhibitor group: Patients with GLDs but without SGLT-2 inhibitors at discharge from the cardiology department. These GLDs included any of (i) insulin, (ii) metformin, (iii) sulfonylurea, (iv) dipeptidyl peptidase 4 inhibitors, (v) glucagon-like peptide-1 receptor agonist (GLP1-ra), (vi) thiazolidinediones, or (vii) other GLDs (acarbose, bromocriptine, miglitol, nateglinide, and repaglinide).

Recording of anthropometric characteristics, clinical variables, and laboratory parameters was performed at baseline (Figure 1 and Appendix A (Supplementary Materials)). The prescription and dose of GLDs at discharge was at the discretion of the treating cardiologist. Follow-up after discharge consisted of 6, 12, and 18 months visits at the outpatient clinic (Figure 1 and Appendix A). During hospitalization, no patient was given SGLT-2 inhibitors.

### 2.4. Study Endpoints

The primary endpoints of this study were safety endpoints, which included (i) adverse events leading to discontinuation of GLDs, (ii) worsening renal function (renal composite outcome defined as sustained decrease of 40% or more in estimated glomerular filtration rate (eGFR)—calculated by means of the Chronic Kidney Disease Epidemiology Collaboration equation [16]—reaching eGFR values below 60 mL per minute per 1.73 m^2^ of body surface area, or new end-stage renal disease (dialysis for at least 30 days, kidney transplantation, or an eGFR of <15 mL per minute per 1.73 m^2^ sustained for at least 30 days), or death from renal causes), (iii) hospitalization for renal cause defined as any hospitalization for acute kidney injury, (iv) hospitalization for hepatic injury, metabolic acidosis, ketoacidosis, and diabetic ketoacidosis.

The secondary efficacy clinical outcomes were (i) death from any cause, (ii) death from cardiovascular cause, (iii) readmissions for cardiovascular cause, and (iv) combined clinical outcome (death or readmission for cardiovascular cause).

### 2.5. Statistical Analysis

Continuous variables were analyzed for a normal distribution with the Kolmogorov–Smirnov test and presented as mean ± standard deviation or as median and interquartile range if a normal distribution was present or not, respectively. Student’s t test or Mann–Whitney’s *U* test was used for comparisons of continuous variables where appropriate.

Categorical variables are expressed as frequencies and percentages. Categorical variables were tested by means of the *Χ*^2^ test or Fisher’s exact test when at least 25 % of values showed an expected cell frequency below 5.

Differences between mean values between discharge and follow-up were evaluated for paired Student’s *t* test.

We performed a propensity-score (PS) matching analyses to ensure baseline balance at initiation of SGLT-2 inhibitors or non-SGLT-2 inhibitors. Variables for PS calculations were age, sex, left ventricular ejection fraction (at index date), cerebrovascular disease (diagnosed before the index date), estimated glomerular filtration rate <60 mL/min/1.73 m^2^ calculated by means of the Chronic Kidney Disease Epidemiology Collaboration equation (at index date), and metformin therapy (at discharge). These variables were selected because they are known potential risk factors of GLD safety and efficacy or because they showed differences between both groups in the univariate analysis. PS was calculated using the logistic regression model, and the nearest-neighbor caliper width of 0.1 multiplied by the standard deviation of the PS distribution was used for matching. SGLT-2 inhibitors patients were matched to non-SGLT-2 inhibitors patients in a 1:1 ratio. Balancing after matching was assessed by standardized differences. A standardized difference greater than 10% indicates a residual imbalance between two groups.

Kaplan–Meier plots were generated to characterize the contour of risk over time for each outcome. Conditional Cox proportional hazards regression based on time to first event was used to estimate hazard ratios (HRs) and 95% confidence intervals (CIs), comparing the treatment effect of SGLT-2 inhibitors against that of non-SGLT-2 inhibitors (reference group) in relation to each study endpoint. When no events were shown in one of the groups, a logistic regression was used instead, to obtain the odds ratio and 95% CIs.

We considered a *p*-value less than 0.05 as statistically significant. We used R Statistical Software (R Core Team, Vienna, Austria 2020) in all data analyses.

## 3. Results

### 3.1. Patients, GLD Regimens, and Follow-Up

Overall, 1763 patients at hospital discharge from the cardiology department were identified during the study period, with 755 eligible discharges (no elective hospitalizations with 3 or more days of inpatient stay) and among which 104 were T2DM patients with at least one or more prescriptions of GLDs at discharge from the cardiology department. Of these, 38 patients (37%) initiated an SGLT-2 inhibitor at hospital discharge from the cardiology department, so, they were classified to the SGLT-2 inhibitors group, and 66 (63%) initiated any GLD non SGLT-2 inhibitor, so, they were classified to the non-SGLT-2 inhibitors group. Moreover, 5 patients (5%) had received SGLT-2 inhibitors treatment before hospital admission in the cardiology department, in all of them, SGLT-2 inhibitors were suspended during hospitalization and, at discharge, at the discretion of their treating cardiologist, 4 patients restarted SGLT-2 so they were assigned to the SGLT-2 inhibitor group and 1 patient, who did not restart SGLT-2 inhibitors, was assigned to the non SGLT-2 inhibitor group. After PS matching, 76 patients were ultimately matched and selected for comparison. Specifically, 38 patients (100% retention of the total eligible) from the SGLT-2 inhibitor group were matched 1:1 with 38 patients from the non-SGLT-2 inhibitor group. Among the SGLT-2 inhibitors group including: 34 patients who had initiated empagliflozin at cardiology discharge, 3 on dapagliflozin, and 1 on canagliflozin at discharge. The flow chart of this study is presented in Figure 1.

Table 1 shows the GLD regimens in the two groups (SGLT-2 inhibitors/non-SGLT-2 inhibitors) before and after propensity-score matching.

Baseline characteristics in the two groups are presented in the Table 2 before and after PS matching. Before matching, compared with non-SGLT-2 inhibitors, patients newly prescribed an SGLT-2 inhibitor therapy at baseline were younger and less frequently had a history of ischemic stroke or chronic kidney disease. In addition, new users of an SGLT-2 inhibitor presented with higher rates of background treatment with metformin. After PS matching, all baseline patient characteristics included in the PS model were well balanced (standardized differences <0.1 for all baseline characteristics after propensity matching), and no significant differences between groups in glycemic profile (glycated hemoglobin level, duration of diabetes, or prior diagnosis/new diagnosis of T2DM) (Table 2) or GLD treatment (Table 1), were found.

Among the matched cohort, the mean age of the patients was 68 ± 11 years, 66% were male, 5% had chronic kidney disease, mean left ventricular ejection fraction was 53%. All patients had T2DM, 37% had a new diagnosis of T2DM during hospitalization and 63% had prior diagnosis of T2DM. The mean glycated hemoglobin level was 7.56% ± 1.6% and the median duration of diabetes was 5.6 ± 0.004 years. At baseline, among the matched cohort overall, 82% of patients were treated with metformin and 22% with insulin. The diagnoses at discharge from the cardiology department were acute coronary syndrome 61%, heart failure 17%, and arrhythmias 9%.

The mean follow-up was 16 ± 2 months for the cohort. No patient was lost to follow-up and all endpoints were available in all patients.

### 3.2. Safety Outcomes

Safety data of SGLT-2 inhibitors are presented in Table 3. The incidence rates of discontinuation of GLDs or adverse events were similar in subjects treated with SGLT-2 inhibitors or non-SGLT-2 inhibitors.

In the SGLT-2 inhibitor group, only 1 patient (3%) presented worsening renal function that forced the temporary suspension of SGLT-2 inhibitors. This patient had started at discharge from the cardiology department, angiotensin-converting enzyme inhibitor and mineralocorticoid receptor antagonist along with SGLT-2 inhibitors. His worsening renal function was managed as an outpatient with rapid resolution after withdrawal of nephrotoxic drugs (angiotensin-converting enzyme inhibitor and mineralocorticoid receptor antagonist); this subsequently allowed the reintroduction of SGLT-2 inhibitors.

No patient death from renal cause, but in non-SGLT-2 inhibitor group, 2 patients (5.3%) were hospitalized for renal cause during follow-up (Table 3).

None of the patients were hospitalized for hepatic injury, metabolic acidosis, ketoacidosis, or diabetic ketoacidosis.

### 3.3. Efficacy Clinical Outcomes

With respect to efficacy, the SGLT-2 inhibitor group resulted in a lower rate of death or readmission for cardiovascular cause (combined clinical outcome) than the non-SGLT-2 inhibitor group (7 (18.42%) vs. 16 (42.11%), respectively; hazard ratio (HR), 0.35; 95% CI, 0.14 to 0.85; *p* = 0.02) (Figure 2 or Figure 3D).

Any-cause death was lower in the SGLT-2 inhibitor group than that in the non-SGLT-2 inhibitor group (0 (0%) vs. 9 (23.68%); HR, 0.79; 95% CI, 0.69 to 0.9; *p* = 0.001) (Figure 2 or Figure 3A). Similar results were obtained for cardiovascular death (0 (0%) vs. 7 (17.42%); HR, 0.83; 95% CI, 0.73 to 0.94; *p* = 0.005) (Figure 2 or Figure 3B). In fact, there were no deaths in the SGLT-2 inhibitor group. Appendix A details each of the patients who died during follow-up with their clinical characteristics and the cause of death.

No significant differences were obtained in readmissions for cardiovascular cause (7 (18.42%) vs. 10 (26.32%); HR, 0.63; 95% CI, 0.24 to 1.60; *p* = 0.349) (Figure 2 or Figure 3C). Although it is true that the number of readmissions for heart failure was higher in the group of non-SGLT-2 inhibitors, this difference was not statistically significant. Appendix A details the clinical efficacy endpoints, stratifying readmissions according to their etiology.

Moreover, a significant increase in survival free of death from any cause (log rank test *p*-value 0.001) or cardiovascular death (long rank test *p*-value 0.004) was found in the SGLT-2 inhibitor group. No differences were found in survival free of cardiovascular readmission for both groups (long rank test *p*-value 0.35). Additionally, a significant increase in survival free of both (death from any cause or cardiovascular readmission) was also found (long rank test *p*-value 0.015).

Detailed efficacy clinical data are presented in Figure 2 and Figure 3.

### 3.4. Other Outcomes

During follow-up, the SGLT-2 inhibitor group presented a significant decrease in glycated hemoglobin (7.8% to 7.1%, *p* = 0.04) and weight (82 to 79 kg, *p* < 0.01). There were no differences in blood pressure (systolic and diastolic) or estimated glomerular filtration rate (Figure 4 and Appendix A).

The non-SGLT-2 inhibitor group showed a similar behavior to that of the SGLT-2 inhibitor group in terms of glycated hemoglobin and blood pressure reductions. However, it did not produce significant changes in weight throughout the follow-up. However, statistically significant reductions in GFR were observed in non-SGLT-2 inhibitor group during follow-up (72 ± 21 vs. 64 ± 25, *p* = 0.02) (Figure 4 and Appendix A).

Among the matched cohorts, after 16 ± 2 months follow-up, the mean of LDL cholesterol was 68 ± 26 mg/dL, the mean of blood pressure was 128/72, and the mean of glycated hemoglobin was 7.2% without differences between groups.

## 4. Discussion

In this cohort study with prospective follow-up, we evaluated the safety and efficacy of the prescription of SGLT-2 inhibitors in T2DM patients at discharge from cardiology hospitalization. As far as we know, this is the first study to analyze this new scenario of SGLT-2 inhibitor prescription on hospital discharge.

The present study shows that the initiation of SGLT-2 inhibitors at discharge from cardiology hospitalization was safe. We saw no evidence, despite focused collection of events, of a higher risk of worsening renal function, hospitalization, or death from renal cause than with non-SGLT-2 inhibitors. Furthermore, during follow-up, patients who initiated SGLT-2 inhibitors at hospital discharge presented a lower rate of cardiovascular death and any-cause death.

### 4.1. SGLT-2 Inhibitors and Cardiovascular Disease

SGLT-2 inhibitors were originally designed as glucose-lowering drugs for glycemic control in T2DM patients [17]. In their large randomized controlled trials in patients with diabetes, SGLT-2 inhibitors consistently reduced renal and cardiovascular events, and HF hospitalization [8,9,10]. Therefore, current guidelines for T2DM management recommend the use of SGLT-2 inhibitors as second-line treatment, following metformin. Even for specific groups of people with T2DM, such as those with multiple risk factors for atherosclerotic disease or a history of cardiovascular events, recently published recommendations suggest using these agents as first-line therapy [6,7], due to evidence supporting their potential to decrease CV risk, particularly at the secondary-prevention level [1,2,3,4,5,11,12]. However, the trials of SGLT-2 inhibitors were performed in stable, outpatient T2DM cases [8,9,10]. Until now, no data is available on the safety and efficacy of SGLT-2 inhibitors at hospital discharge [13].

T2DM patients who are discharged from the cardiology department have multiple risk factors for atherosclerotic disease or just suffered a cardiovascular event or acute HF, and their risk of death and readmission is greater during the first year after discharge [18,19]. Additionally, the prevalence of T2DM among hospitalized patients in cardiology is very high (30%) [20], and even a 16% of new diagnosis of T2DM could be detected if an adequate screening was requested (according to the guidelines’ recommendation [6,7]) during cardiology hospitalization [20,21]. Therefore, cardiology hospitalization should be considered as an opportunity for the detection of T2DM and for the initiation of antidiabetic drugs with cardiovascular benefits [13].

Actual studies with contemporary multicenter cohorts [14,15,16,17,18,19,20,21,22], found suboptimal rates of use of secondary prevention therapies in high-risk patients with diabetes and atherosclerotic cardiovascular disease particularly with high-intensity statins and glucose-lowering therapies with proven cardiovascular benefit (such as SGLT-2 inhibitors). Given the high cardiovascular event rates in this patient population, improving the use of these guideline-recommended therapies [6,7] is an important potential opportunity to improve care and, in turn, reduce the risk of recurrent atherosclerotic cardiovascular disease events, heart failure hospitalizations, and cardiovascular mortality. 

The same happened with the angiotensin receptor–neprilysin inhibitors, in that, despite strong clinical evidence and the clear recommendations of the guidelines, the initial prescription of this drug was low [23]. Furthermore, the publication of the study on safety and efficacy at hospital discharge (TRANSITION Study [24]) helped to increase prescription by analyzing a new clinical scenario for its use. For this reason, our study tries to shed light on this new scenario by showing that SGLT-2 inhibitors at hospital discharge are safe and with clinical benefit. However, others clinical trials, to study safety and efficacy of these SGLT-2 inhibitors and GLP1a in this context are greatly needed, such as the SOLOIST-WHF (NCT03521934), which will study the effect of sotaglifozin in this context.

### 4.2. Safety of Prescription of SGLT-2 Inhibitors at Hospital Discharge

There are several key findings from this study. With respect to the safety of SGLT-2 inhibitors at discharge from the cardiology department. Only 1 patient (3%) presented worsening renal function that forced the temporary suspension of SGLT-2 inhibitor. It is necessary to clarify that at hospital discharge this patient started angiotensin-converting enzyme inhibitor and mineralocorticoid receptor antagonist along with SGLT-2 inhibitor. Therefore, it is possible that his worsening renal function was related to the use of these drugs and not to the prescription of the SGLT-2 inhibitor. In the present study, 17% of included patients were discharged after acute heart failure treated with intravenous loop diuretics, and 20% were cotreated with oral loop diuretics during follow-up. There was no patient death from renal cause, no hospitalization for renal cause, or hepatic injury, metabolic acidosis, ketoacidosis, or diabetic ketoacidosis. Thus, SGLT-2 inhibitors are safe and well tolerated in this scenario. These data are consistent with the results of large, randomized controlled trials with SGLT-2 inhibitors in stable outpatients [8,9,10], and a recent study of empaglifozin in hospitalized patients admitted for heart failure also shows that empagliflozin was safe and well tolerated with less adverse events than placebo without more frequent worsening of renal function or renal adverse events [25].

### 4.3. Morbidity and Mortality Impact of SGLT-2 Inhibitors

SGLT-2 inhibitors did result in a lower rate of the any-cause death than non-SGLT-2 inhibitors. Moreover, we did not observe any death in the SGLT-2 inhibitor group. However, these data should be interpreted with caution because this study was not powered and not designed to show an effect on clinical endpoints, and the number of events was very low. The reduction in clinical endpoints is consistent with previous morbidity and mortality benefits for SGLT-2 inhibitors in T2DM trials [8,9,10] and meta-analyses [1,2,3,4]. Furthermore, our results in combined clinical outcome and each of the secondary efficacy clinical outcomes are in line. Therefore, this study suggests that the prescription of SGLT-2 inhibitors initiated at hospital discharge could reduce the mortality in a T2DM high-risk population. It is interesting to note that the two groups in the present study presented excellent control of cardiovascular risk factors with mean LDL levels <70 mg/dL, BP levels <135/85, and glycated hemoglobin of 7.2% without differences between the groups. This supports the suspicion that the beneficial effects of SGLT-2 inhibitors are not only secondary to better control of blood pressure, weight, and blood glucose.

### 4.4. Limitations

This study has several limitations. First and foremost, this study is limited by the number of patients from a single center and should be considered as a pilot study in terms of efficacy outcomes. Secondly, although results were adjusted for PS matching, the prescription of GLDs at discharge was at the discretion of the treating cardiologist without randomization. Finally, there was no standardized protocol for at-discharge prescriptions, which means individual differences in the treatment of these patients may have impacted the results.

## 5. Conclusions

SGLT-2 inhibitor prescription at hospital discharge in heart disease T2DM patients was safe and well tolerated, moreover it was associated with a reduction in all-cause and cardiovascular deaths during follow-up. Larger, randomized clinical trials with SGLT-2 inhibitors are greatly needed to further study the possible beneficial role of SGLT-2 inhibitors in this clinical scenario.

## Figures and Tables

**Figure 1 jcm-09-02600-f001:**
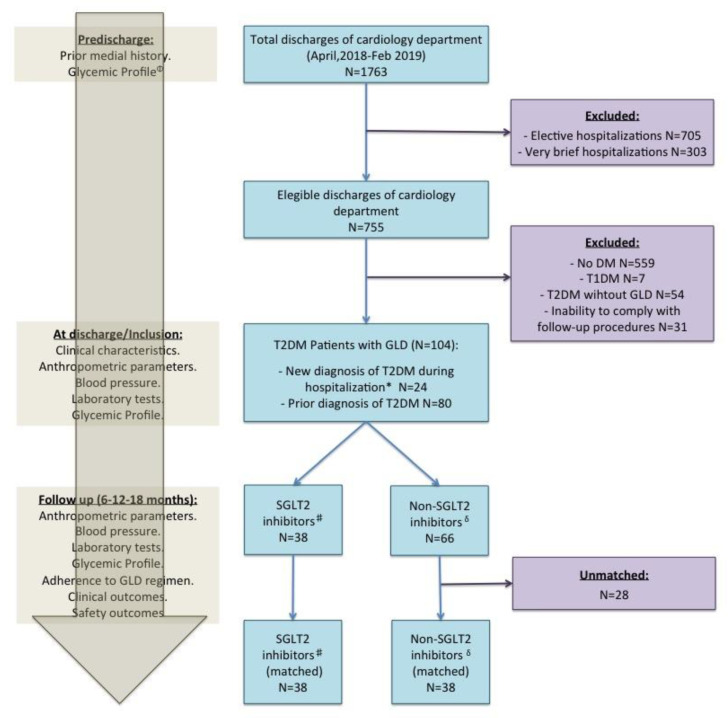
Flow chart of study. Left panel show study procedures. ^#^ Sodium-glucose cotransporter-2 (SGLT-2) inhibitors: Patients who had initiated SGLT-2 inhibitors at discharge from the cardiology department. ^δ^ Non-SGLT-2 inhibitors: Patients with glucose-lowering drugs (GLD) but without SGLT-2 inhibitors at discharge from the cardiology department. * New diagnosis of type 2 diabetes mellitus (T2DM) during hospitalization according to the American Diabetes Association criteria (19). ^Φ^ Glycemic profile: Glycated hemoglobin and fasting plasma glucose. DM: diabetes mellitus; FPG: fasting plasma glucose; GLD: glucose-lowering drugs; SGLT-2: sodium-glucose cotransporter-2; T1DM: type 1 diabetes mellitus; T2DM: type 2 diabetes mellitus.

**Figure 2 jcm-09-02600-f002:**
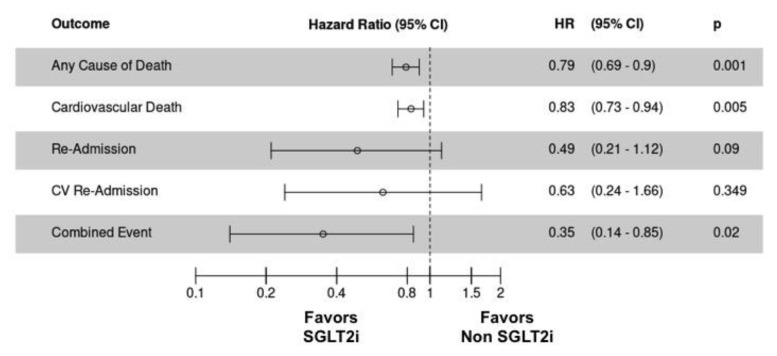
Efficacy clinical outcomes for patients in the propensity-score matched cohort by groups (SGLT-2 inhibitors/non-SGLT-2 inhibitors). CI: confidence interval; CV: cardiovascular; HR: hazard ratio; SGLT-2i: sodium-glucose cotransporter-2 inhibitors.

**Figure 3 jcm-09-02600-f003:**
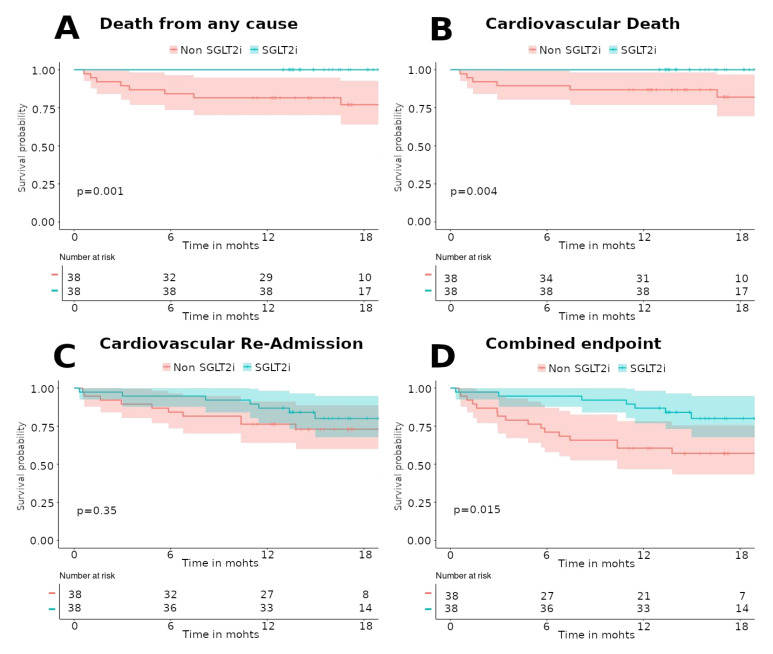
Kaplan–Meier survival analysis for death from any cause (**A**), cardiovascular death (**B**), cardiovascular readmission (**C**), and combined endpoint (death or cardiovascular readmission), (**D**) for patients in the propensity-score matched cohort by groups (SGLT-2 inhibitors/non-SGLT-2 inhibitors). Statistically significant differences were found for death from any cause, cardiovascular death, and combined endpoint. No differences were found for cardiovascular readmission. SGLT-2i: sodium-glucose cotransporter-2 inhibitors.

**Figure 4 jcm-09-02600-f004:**
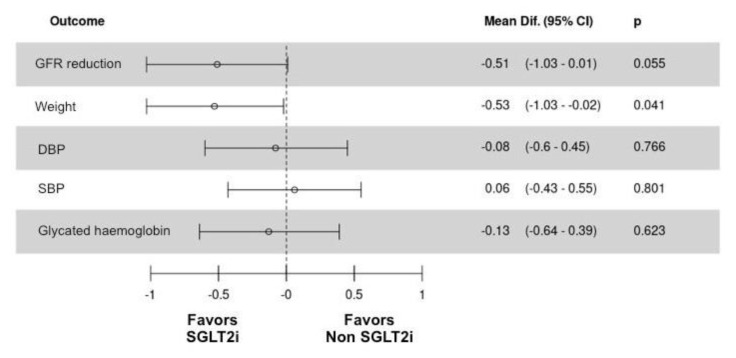
Z-score presentation of estimated glomerular filtration rate reduction, weight, blood pressure (systolic and diastolic), and glycated hemoglobin for patients in the propensity-score matched cohort by groups (SGLT-2 inhibitors/non-SGLT-2 inhibitors). CI: confidence interval; DBP: diastolic blood pressure; GFR: glomerular filtration rate; SGLT-2i: sodium-glucose cotransporter-2 inhibitors; SBP: systolic blood pressure.

**Table 1 jcm-09-02600-t001:** Glucose-lowering drug regimens of patients in both groups before and after propensity-score matching.

Glucose-Lowering Drugs	Unmatched			Propensity-Score Matched		
	SGLT-2 Inhibitors (*n* = 38)	Non-SGLT-2 Inhibitors (*n* = 66)	*p*-Value	SGLT-2 Inhibitors (*n* = 38)	Non-SGLT-2 Inhibitors (*n* = 38)	*p*-Value
Metformin	35 (92%)	34 (52%)	<0.001	35 (92%)	31 (82%)	0.309
Dipeptidyl peptidase 4 inhibitors	8 (21%)	18 (27%)	0.638	8 (21%)	8 (21%)	1
Sulfonylurea	2 (5%)	3 (5%)	1	2 (5%)	2 (5%)	1
Insulin	9 (24%)	21 (32%)	0.511	9 (24%)	7 (18%)	0.778
Sodium-glucose cotransporter 2 inhibitors	38 (100%)	0	–	38 (100%)	0	–
Glucagon-like peptide 1 receptor agonist	3 (8%)	1 (2%)	0.271	3 (8%)	1 (3%)	0.615

SGLT-2: sodium-glucose cotransporter-2; “–“: not applicable.

**Table 2 jcm-09-02600-t002:** Baseline characteristics in both groups (SGLT-2 inhibitors/non-SGLT-2 inhibitors) before and after propensity-score matching.

Characteristics	Unmatched					Propensity-Score Matched				
	SGLT-2 Inhibitors (*n* = 38)		Non-SGLT-2 Inhibitors (*n* = 66)			SGLT-2 Inhibitors (*n* = 38)		Non-SGLT-2 Inhibitors (*n* = 38)		
	*n*/Mean	%/SD	*n*/Mean	%/SD	*p*-Value	*n*/Mean	%/SD	*n*/Mean	%/SD	*p*-Value
Prior history										
Age (years)	65.7	10.6	74	11.3	<0.001	65.7	10.6	69.4	11.4	0.152
Male sex *n* (%)	25	65.8	43	63.2	0.947	25	65.8	25	65.8	1
Smoke *n* (%)	6	15.8	6	9.1	0.303	6	15.8	4	10.5	0.197
Hypertension *n* (%)	21	55.3	36	63.2	0.944	21	55.3	16	42.1	0.359
History of chronic kidney disease ^Ψ^ *n* (%)	2	5.3	22	33.3	0.001	2	5.3	3	7.89	1
History of cerebrovascular disease *n* (%)	0	0	8	12.1	0.025	0	0	0	0	1
History of heart failure *n* (%)	5	13.1	15	22.7	0.233	5	13.1	7	18.4	0.753
History of coronary artery disease *n* (%)	11	28.9	18	27.3	0.858	11	28.9	10	26.32	1
Discharge diagnosis										
STEMI *n* (%)	6	15.8	4	6.1	0.105	6	15.8	2	5.3	0.135
NSTEMI *n* (%)	8	21.1	21	31.8	0.238	8	21.1	13	34.2	0.200
Unstable Angina *n* (%)	9	23.7	10	15.2	0.278	9	23.7	9	23.7	1
Stable heart disease *n* (%)	3	7.9	4	6.1	0.719	3	7.9	2	5.3	0.644
Heart failure *n* (%)	7	18.4	15	22.7	0.605	7	18.4	15	22.7	0.605
Arrhythmia *n* (%)	4	10.5	8	12.1	0.806	4	10.5	8	12.1	0.806
Others *n* (%)	1	2.6	3	4.5	0.625	1	2.6	3	4.5	0.625
Anthropometrical parameters, laboratory tests, and glycemic profile (at inclusion)										
Systolic blood pressure (mmHg)	124.8	16.3	127.3	16.5	0.453	124.789	16.28	125.189	13.737	0.909
Diastolic blood pressure (mmHg)	69.9	8.9	65.8	12.8	0.057	69.947	8.88	68.622	13.149	0.612
Body mass index ^Φ^ (%)	29.091	4.485	28.321	5.327	0.453	29.091	4.485	28.72	6.081	0.776
Fasting plasma glucose (mg/dL)	137.447	49.465	145.97	64.839	0.453	137.447	49.465	147.632	66.039	0.449
Glycated hemoglobin (%)	7.787	1.596	7.285	1.554	0.124	7.787	1.596	7.286	1.619	0.182
eGFR ^φ^ (mL/min/1.73 m^2^)	77.343	18.148	58.19	27.425	<0.001	77.343	18.148	71.886	21.351	0.253
LDL-cholesterol (mg/dL)	84.211	36.1	84.523	38.861	0.967	84.211	36.1	89.622	44.013	0.563
LVEF (%)	52.71	11.536	53.861	12.856	0.667	52.71	11.536	51.848	12.592	0.776
Duration of T2DM (months)	66.9	0.2	66.9	0.3	0.977	66.9	0.2	67.0	0.2	0.261
Cardiovascular therapies										
Antiplatelet agents *n* (%)	26	68.4	46	69.7	0.892	26	68.42	27	71.05	1
Anticoagulants *n* (%)	6	15.8	12	18.2	0.756	6	15.79	5	13.16	1
ACE inhibitor or ARB *n* (%)	26	68.4	25	37.9	0.003	26	68.42	15	39.47	0.021
MRA *n* (%)	1	2.6	4	6.1	0.431	1	2.63	4	10.53	0.358
Beta-blocker *n* (%)	25	65.8	44	66.7	0.927	25	65.79	26	68.42	1
Statins *n* (%)	29	76.3	48	72.7	0.688	29	76.32	27	71.05	0.794
Ezetimibe *n* (%)	3	7.9	12	18.2	0.15	3	7.89	8	21.05	0.191
Loop diuretics *n* (%)	6	15.8	25	37.9	0.018	6	15.79	11	28.95	0.271

^Ψ^ History of chronic kidney disease was defined by estimated glomerular filtration rate (eGFR) >60 mL/min/1.73 m². ^Φ^ Body mass index is the weight in kilograms divided by the square of the height in meters. ^φ^ Estimated glomerular filtration rate (eGFR): calculated by means of the Chronic Kidney Disease Epidemiology Collaboration equation. ACE: angiotensin-converting enzyme; ARB: angiotensin-receptor blocker; eGFR: estimated glomerular filtration rate; HF: heart failure; LDL: low-density lipoprotein; LVEF: left ventricular ejection fraction; MRA: mineralocorticoid receptor antagonist; NSTEMI: non-ST-segment elevation myocardial infarction; SD: standard deviation; SGLT-2: sodium-glucose cotransporter-2; STEMI: ST elevation myocardial infraction.

**Table 3 jcm-09-02600-t003:** Safety outcomes for patients in the propensity-score matched cohort by groups (SGLT-2 inhibitors/non-SGLT-2 inhibitors).

Safety Outcome	SGLT-2 Inhibitors (*n* = 38)	Non-SGLT-2 Inhibitors (*n* = 38)	*p*-Value
Discontinuation of SGLT-2 inhibitors	1 (3%)	–	–
Discontinuation of other GLDs	10 (26%)	10 (26%)	1
Worsening renal function	1 (2.6%)	2 (5.3%)	0.94
Death from renal cause	0	0	1
Hospitalization for renal cause	0	2 (5.3%)	0.16
Hospitalization for hepatic injury, metabolic acidosis, ketoacidosis, or diabetic ketoacidosis.	0	0	1

GLD: glucose-lowering drugs; SGLT-2: sodium-glucose cotransporter-2; “–“: not applicable.

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
