# Peer review of "Sodium-Glucose Cotransporter-2 Inhibitors at Discharge from Cardiology Hospitalization Department: Decoding A New Clinical Scenario"

_jcm, 2020, doi:10.3390/jcm9082600_

Round 1

Reviewer 1 Report

It is a very interesting study that might help increase the prescription of SGLT-2 inhibitors in the post-acute scenario. The main limitations are the small size and that it is not a randomized study. However, the authors have tried to minimize the risk of bias.

I assume that all patients included in the study were de novo SGLT2 inhibitors but I guess that in the whole cohort some patients were under this treatment. In Figure 1 I cannot see the number of patients with previous use of iSLGT2 that were excluded from the study.

Why did patients with new diagnosis of T2DM receive treatment with iSGLT2 as first line treatment? Is this the current practice in our center?

The study endpoints are not clear: worsening renal failure was defined as a decrease of 40% or more in eGFR only if eGFR was then less than 60 mL/min/1.73m2; or was it <40% eGFR or eGFR<60 mL/min. What was considered a hospitalization for renal cause? Where is the need for hemodialysis in this classification?

Given the small sample size it might be difficult to draw any conclusion, but have the authors noticed whether a specific primary diagnosis was associated with better outcomes after iSGLT2 initiation? For instance, compared to non-iSLGT2 users, had HF patients better prognosis than patients with acute myocardial infarction?

What were the main causes of CV rehospitalization? Ischemic heart disease or heart failure?

Could other medication prescribed at discharged have affected the results (i.e. statins, DAPT, ACE inhibitors and beta blockers in HF with reduced ejection fraction…). Cardiovascular risk factors were extremely well controlled. Unfortunately, this is not what we see in clinical practice. Do the authors think that their results could be replicated in less optimally controlled patients?

Author Response

Response to Reviewer 1 Comments:

We want to thank you for your interest in our work and for the interesting comments proposed, as we believe that these will help improve the quality and scientific rigor of our work so that it can be published at the JCM.

In the following lines, we detail each of the comments proposed by the reviewer and reference the changes made in the new adapted manuscript.

Point 1: I assume that all patients included in the study were de novo SGLT2 inhibitors but I guess that in the whole cohort some patients were under this treatment. In Figure 1 I cannot see the number of patients with previous use of iSLGT2 that were excluded from the study.

Response 1: In our study, 5 patients (5%) had received SGLT-2 inhibitors treatment before hospital admission in cardiology department, in all of them SGLT-2 inhibitors was suspended during hospitalization and, at discharge, at the discretion of their treating cardiologist (unrelated to investigators), 4 patients restarted SGLT-2 inhibitors so they were assigned to the SGLT-2 inhibitor group and 1 patient, who did not restart SGLT-2 inhibitors, was assigned to the non SGLT-2 inhibitor group. None of these 5 patients was excluded from the study, because none of them were treated with SGLT-2 inhibitors during hospitalization and were subsequently assigned to one or the other group depending on whether or not they had restarted SGLT-2 inhibitors at the time of hospital discharge. The main aim of the present prospective study is to assess the safety of SGLT-2 inhibitors when prescribed at hospital discharge to T2DM patients after a non-elective admission to the Cardiology Department, so we consider that these patients meet the inclusion criteria, without meeting the exclusion criteria and therefore should not be excluded.

Although, we totally agree with the reviewer, that in the previous manuscript, these patients are not explicitly referred to, so in the new manuscript we have incorporated in section 3.1 “Patients, GLD regimens and follow-up ”(Lines 161-165), a new paragraph explaining these 5 patients in detail. This paragraph is attached below:

“5 patients (5%) had received SGLT-2 inhibitors treatment before hospital admission in cardiology department, in all of them SGLT-2 inhibitors was suspended during hospitalization and, at discharge at the discretion of their treating cardiologist, 4 patients restarted SGLT2 so they were assigned to the SGLT-2 inhibitor group and 1 patient, who did not restart SGLT-2 inhibitors, was assigned to the non SGLT-2 inhibitor group.“

Point 2: Why did patients with new diagnosis of T2DM receive treatment with iSGLT2 as first line treatment? Is this the current practice in our center?

Response 2: According to the current European clinical practice guidelines [Cosentino, F. et al. 2019 ESC Guidelines on diabetes, pre-diabetes, and cardiovascular diseases developed in collaboration with the EASD. Eur Heart J. 2020, 41, 255–323] and the consensus documents of the Spanish cardiology society, the first-line treatment in patients with a cardiovascular event are SGLT-2 inhibitors or GLP-1 agonist. Specifically, the cohort of our study are patients hospitalizated in cardiology department, therefore all have developed a clinical cardiovascular event. Therefore, it is standard practice in our center that if there are no contraindications, patients after the diagnosis of DM2 are treated with a combination of Metformin + SGLT-2i or Metformin + GLP-1a.

Point 3: The study endpoints are not clear: worsening renal failure was defined as a decrease of 40% or more in eGFR only if eGFR was then less than 60 mL/min/1.73m2; or was it <40% eGFR or eGFR<60 mL/min. What was considered a hospitalization for renal cause? Where is the need for hemodialysis in this classification?

Response 3: Worsening renal function was defined as a reduction of ≥40% in the baseline GFR (reaching the final GFR <60 ml/min/1.73m2). The need for hemodyalisis was included as a new end-stage renal disease. And renal hospitalization was defined as any hospitalization of at least 24 hours due to renal exacerbation.

We totally agree with the reviewer, in that we have not been able to unequivocally express these concepts, so in the new manuscript we have replaced the paragraph of section 2.4 "study endpoints" (lines 98-106) with a new paragraph with the more detailed explanation that I attach below:

“The primary endpoints of this study were safety endpoints, which included (i) adverse events leading to discontinuation of GLD, (ii) worsening renal function (renal composite outcome defined as sustained decrease of 40% or more in estimated glomerular filtration rate (eGFR) —calculated by means of the Chronic Kidney Disease Epidemiology Collaboration equation [16]— reaching eGFR values below 60 ml per minute per 1.73 m2 of body surface area, or new end-stage renal disease (dialysis for at least 30 days, kidney transplantation, or an eGFR of <15 ml per minute per 1.73 m2 sustained for at least 30 days), or death from renal causes), (iii) hospitalization for renal cause defined as any hospitalization for acute kidney injury, (iv) hospitalization for hepatic injury, metabolic acidosis, ketoacidosis and diabetic ketoacidosis.”

Point 4: Given the small sample size it might be difficult to draw any conclusion, but have the authors noticed whether a specific primary diagnosis was associated with better outcomes after iSGLT2 initiation? For instance, compared to non-iSLGT2 users, had HF patients better prognosis than patients with acute myocardial infarction?

Response 4: Certainly, this reviewer's comment seems very relevant to us. SGLT- 2 inhibitors are known to reduce HF in their pivotal studies. In our study, 65% of the matched cohort are patients with an acute myocardial infarction and 20% with HF. Statistical analysis of the subgroup of patients with and without HF was performed for each of the efficacy endpoints, but probably due to the small sample size (not enhanced for this type of subgroup analysis), no differences were found in any of the endpoints studied. For this reason, we considered not including it in the manuscript to avoid straying from the objectives of the study and not transmitting the wrong message in this regard.

Point 5: That were the main causes of CV rehospitalization? Ischemic heart disease or heart failure?

Response 5: Again, the comment proposed by the reviewer seems very appropriate to us. The association of SGLT-2 inhibitors with the HF reduction is much greater than with Ischemic heart disease (IHD) reduction in the actual evidence. In our study in the SGLT-2 inhibitor group: 7 patients readmission for CV causes (Of these, 86% were due to Acute coronary syndrome (ACS) and only 14% for HF); unlike the non-SGLT-2 inhibitor group where 10 patients re-admission for CV causes (60 % by ACS and 40% by HF). Therefore, in light of these dates, a trend towards a lower number of readmissions for HF in the SGLT-2 inhibitor group is observed, although our small sample size does not allow us to reach statistical significance in this section (p = 0.3). It is important to point out that the primary and main objective of our study was to assess the safety of SGLT-2 inhibitors when prescribed at hospital discharge to T2DM patients after a non-elective admission to the Cardiology Department, and for this purpose its sample size was designed.

We agree with the reviewer that we find it interesting to report this trend in readmissions for HF in the manuscript, so the following table S3 has been added to the supplementary material and in the manuscript results section 3.3 (lines 242-245), in which each of the clinical efficacy endpoints are specified in detail and with the CV re-admission subgroups (HF and non-HF). Below I attach text added to the new manuscript (lines 242-245) and supplementary table.

“Although it is true that the number of readmissions for heart failure was higher in the group of non-SGLT2 inhibitors, this difference was not statistically significant. Table S3 of the supplementary material details the clinical efficacy endpoints, stratifying readmissions according to their etiology.”

Table S3: Efficacy clinical outcomes for patients in the propensity-score matched cohort by groups (SGLT-2 inhibitors/non-SGLT-2 inhibitors).

Efficacy clinical outcome

SGLT-2 inhibitors (N=38)

Non-SGLT-2 inhibitors (N=38)

P-value

Any Cause of Death

0

9 (24%)

0.001

Cardiovascular Death

0

7 (18%)

0.005

·       Non Heart Failure

   0

   4 (11%)

0.04

·       Heart Failure

   0

   3 (8%)

0.077

Re-Admission (all causes)

9 (24%)

15 (39%)

0.09

Cardiovascular Re-Admission

7 (18%)

10 (26%)

0.349

·       Hospitalization for Myocardial Infarction or Ischemic Stroke

     6 (16%)

     6 (16%)

1

·       Hospitalization for Heart Failure

     1 (2%)

     4 (11%)

0.358

Re-Admission for renal cause

0

2 (5%)

0.16

Combined clinical outcome*

7 (18%)

16 (42%)

0.02

*Combined clinical outcome was defined to rate of death or readmission for cardiovascular cause.

CV: Cardiovascular; SGLT2: sodium-glucose cotransporter-2.

Point 6: Could other medication prescribed at discharged have affected the results (i.e. statins, DAPT, ACE inhibitors and beta blockers in HF with reduced ejection fraction…).

Response 6: In the final matched cohort after Propensity score matching, only the prescription of ACE inhibitors showed differences between the two groups, but not in the rest of the treatments (statins, Bblockers, DAPT...). Regarding the differences found in the prescription of ACE inhibitors in the 2 groups, if we extract patients with LVEF<40% (9 patients 12% of the matched cohort), no significant differences were observed in the prescription of ACE inhibitors (62 vs 53% ), or B-blockers (71 vs. 68%) in the SGLT-2 inhibitor and Non-SGLT-2 inhibitor groups respectively. Therefore, we believe that a confounding effect cannot be attributed to the other cardiological treatments in relation to the analyzed endpoints and the results obtained in our study.

Point 7: Cardiovascular risk factors were extremely well controlled. Unfortunately, this is not what we see in clinical practice. Do the authors think that their results could be replicated in less optimally controlled patients?

Response 7: It is true that, unlike what we found in registries of our environment such as EUROASPIRE V, in our study an excellent control of CVRF during follow-up stands out, and this control was equally good in both study groups (no differences between they in LDL, TAS, TAD, HbA1c or weight). We believe that this aspect not only does not invalidate the favorable results found in our work, but also adds additional value to this work. The control of CVRF in DM2 patients is known to reduce clinical events and mortality. However, despite this control, clinical events persist in many patients with T2DM, which has been called “residual risk” in recent years. Although our main aim was safety of the use of SGLT-2 inhibitors in a new setting (discharge from cardiology department). The benefits found in our study in terms of safety and event reduction cannot be associated with CVRF control, since in both groups they were controlled equally. Therefore, there seems to be a CV protective effect associated with SGLT-2 inhibitors beyond the control of CVRF, and that associated with its effect on BP, weight and HbA1c. Thus, if we transfer our results to a population with poorer CVRF control, it is likely that the benefit of SGLT-2 inhibitors is even greater than that found in our study, since to the protective CV effect objectified in our work, with the effect on the CVRFs (BP, weight, HbA1c) by SGLT-2 inhibitors. So, based on the previously commented, We are convinced that the fact that CVRFs are excellently controlled in our study gives additional value to our results.

The new manuscript with the changes made with the Microsoft Word "Track Changes" function is attached as a word file named (jcm-874861 postreview).

Once these major revisions changes are accepted, we will send the final manuscript to a professional English editing service MDPI to improve the English writing and grammar that provide the manuscript with a better reading and higher linguistic quality.

Thank you again for the brilliant review that you have carried out of our study, since it has greatly helped to improve its quality and clairvoyance, for the first time bringing light to a new clinical scenario such as the prescription of SGLT-2 inhibitors at hospital discharge in the post-acute scenario.

Reviewer 2 Report

This is an interesting study with a problematic design.

The authors identified 104 T2DM patients discharged of Cardiology department with glucose lowering Tx, (of a total of 755 eligible) of whom 38 were started with SGLT2 inhibitor. These patients were healthier than those in whome SGLT2i was not initiated: much less renal failure, less cerebrovascular disease and somewhat less heart failure. The differences were mostly resolved by matching  except for ACE inhibitors which were more commonly used in SGLT2i group. Table 1 suggests that the group with SGLT2i initially had a higher HBA1C and were more intensively treated also this difference was not large.

In this observational study  it should be  made clear  how was the decision was made on starting SGLT2i. Also unclear is if there were any patients who already had SGLT2i on admission.

The paper may be improved if these data and more details on follow up organization are provided.

The therapy is reported to be safe, in this cohort. The authors ignore the trend for worsening renal function in SGLT2i group (Fig 4) what is quite surprizing if one trusts the literature.  

Than the clinical outcome which the authors try to dawnplay: a dramatic reduction in mortality  with a study drug, 0/38 vs. 9/38 within less than 2 years. Most of it happened within less that 6 months. This is too good to be true. A fantastic reduction of mortality is not realistic and suggests an inherent bias in organization and data analysis of this non-randomized study. I think matching failed or did not account for something. How many in each group had cancer? Or amyloidosis?

What is also amazing is that there for no difference in hospitalizations. What were the causes of death? Do SGLT2i prevent sudden death? 

Round 2

Reviewer 1 Report

The authors have successfully answered all my comments.

Reviewer 2 Report

I think the paper is much better now. The necessary details are provided and the authors appropriately point to apparent limitations. I think that in the current form  it makes a valuable and a well deserved publication.